# The Effect of Olfactory Inhalation on KPGA Golfers’ Putting Performance, Postural Stability and Heart Rate

**DOI:** 10.3390/ijerph191912666

**Published:** 2022-10-03

**Authors:** Hyoyeon Ahn, Jihyun Ko

**Affiliations:** 1Department of Physical Education, College of Education, Seoul National University, Seoul 08826, Korea; 2School of Sport Science, College of Sports and Arts, Hanyang University, Ansan 15588, Korea

**Keywords:** postural stability, golf putting performance, arousal regulation, olfactory inhalation, center of pressure (CoP), heart rate (HR)

## Abstract

Some athletes utilize olfactory inhalation treatments using ammonia salt and aromatic oils to attain their peak performance or for physical and psychological relaxation. However, there is still a lack of clear evidence on olfactory inhalation treatment and scent types via precise experiments, and there is no research regarding fine motor control performance in activities such as golf putting. Thus, the purpose of this study was to examine the effects of various olfactory inhalations (lavender, citrus, and ammonia) on professional golfers’ 3-meter putting performance (percentage of success), postural stability (CoP area), and heart rate (HR). In order to examine the effects of olfactory treatment on actual automated task performance, ten professional golfers were recruited for the putting task experiment. During the putting task, a biometric shirt was utilized to record the HR changes, and a force plate was used to measure changes in the CoP area. The results were as follows. First, the olfactory inhalation treatment inhibited the putting performance (no inhalation: 68.75%; lavender: 51.25%; citrus: 40.00%; ammonia: 52.50%); however, no statistically significant difference was found (*p* = 0.115). Second, the olfactory inhalation treatment inhibited postural stability while putting; it had a partially statistically significant lower value (address: *p* = 0.000; downswing: *p* = 0.035; total putting section: *p* = 0.047). Third, the olfactory inhalation treatment decreased the HR during putting; however, there was no statistically significant difference between groups (address: *p* = 0.838; putting: *p* = 0.878; total: *p* = 0.666). This study implies that olfactory inhalation affects putting performance, postural stability, and HR. The effect size results for the olfactory treatment in the CoP area during the putting task (address: *η*^2^ = 0.524; downswing: *η*^2^ = 0.349; total putting section: *η*^2^ = 0.298) suggest that arousal regulation through olfactory inhalation may have negative effects on dynamic postural stability in static tasks such as golf putting, showing the direction of its useful application for athletes in sports.

## 1. Introduction

In the game of golf, putting is an important skill that determines the score at the end of every hole. Golf putting performance requires fine motor control and the green-reading ability to grasp the quality and slope of the green toward each hole cup. As a result, it has been reported that professional golfers focus on their short game for 70% of their training to achieve a good score [1]. Players’ self-evaluation and their control of functional conditions during competitions is an important factor in determining the outcome of various sports events; it is even more important in events that require high accuracy and control, such as golf [2]. Accordingly, research that explores and verifies the factors impacting putting performance has been continuously reported.

One of the most important factors in putting performance is dynamic postural stability. Putting requires precise body movements and low weight displacement, involving the center of mass (CoM) and center of pressure (CoP) [3]. Hence, this suggests the need for a concept that differs from static stability. In the context of a putting task, a low CoP area [4,5], low anterior–posterior (AP) pressure center movement [6], and low left–right direction (medial–lateral, ML) [3] have been shown to help improve putting performance. To confirm the changes in dynamic stability in tasks such as putting, it is necessary to apply the research to professional tour golfers who have trained and honed their putting skills and automation of movement.

Another factor affecting putting performance is the putting position, that is, the distance from the hole cup [7,8]. Pelz (2000) found that PGA (Professional Golfer’ Association) players were also affected by putting distance. It was confirmed that the success rate is lowered to 25% at a distance of 10 ft (3.05 m) [9]. Therefore, in this research experiment, the distance was set to 3 m to control the difficulty and to ensure the participants’ concentration during the putting task.

Athletes become aware of the importance of competitive anxiety in sports situations. Competitive anxiety consists of cognitive and physical anxiety, and it is known that optimal anxiety and arousal levels will help to achieve peak performance. Therefore, during the game, players try to control their psychophysiological stability through techniques such as deep breathing and muscle relaxation [10], and practice inducing psychological relaxation through self-talk, cue words, imagery, etc. [11,12,13]. The optimal anxiety and arousal levels are different depending on the sports event and the level of difficulty. Hanin (2000) emphasized that the optimal zone should be considered rather than focusing on a specific level (point) and suggest the use of the individual zone of optimal functioning (IZOF) theory for individual differences [14]. According to various previous theories and hypotheses related to competitive anxiety, golf putting performance is classified as a relatively stable skill. However, putting is a technique that requires fine motor patterns that are different from shooting and archery. Therefore, this study aims to explore the optimal levels of anxiety and arousal for performing fine motor control techniques, such as golf putting, and to investigate the level of arousal by measuring changes in heart rate (HR).

The stimulus response through the olfactory nerve, one of the body’s five senses, has been reported as a component of the nervous system that responds immediately without going through a sensory process, unlike other sensory organs [15,16]. To improve their performance, powerlifters, American footballers, and ice-hockey players use ammonia salt inhalants during or just before sporting events [17,18]. It has already been suggested through previous studies that various scents of olfactory stimulation affect cognitive, behavioral, emotional, and physiological variables [19]. Besides ammonia, lavender and citrus are the most commonly used aromatherapeutic treatments [20]. Lavender inhalation was found to be effective for reducing anxiety [21], autonomic nervous system regulation [22], ensuring good quality of sleep [23], and to be negative for static postural stability [24]. Additionally, citrus inhalation has been reported to help with energy perception, positive feelings, and static postural stability, and has the same anti-anxiety effects as lavender inhalation [24,25,26,27].

Recently, many studies have been conducted to explain the relationship between actual performance and olfactory inhalation treatment, including both general and handgrip strength improvements, Sargent jumps, and cardiovascular enhancements, as well as isometric mid-thigh pull velocity augmentation [19,28,29,30,31]. However, there is still a lack of approaches considering the optimal level of anxiety and arousal in sports situations or performance in demonstrations of skill. Although the study by Ahn, Pathak, Panday, and Kwon (2018) reported that lavender inhalation has a negative effect on static stability (CoP area) [24], dynamic stability should be considered in techniques such as golf putting. Therefore, in this study, we look at the types of scents (odors) that have a positive effect on golf putting performance to ascertain whether olfactory stimulation helps with putting performance.

Therefore, the purpose of this study is to examine the effect of arousal regulation via olfactory inhalation on performance, CoP area, and HR changes during golf putting. To examine the relationship between olfactory inhalation and dynamic stability through precise experimentation, this study was conducted with a focus on professional golfers. The null hypotheses of this research are as follows:

First, when performing the putting task, olfactory inhalation treatment will not affect performance, and there are no differences depending on the types of scents.

Second, when performing the putting task, olfactory inhalation treatment will not affect the changes in the CoP area, and there are no differences depending on the types of scents.

Third, when performing the putting task, olfactory inhalation treatment will not affect the changes in HR, and there are no differences depending on the type of scents.

## 2. Methods

### 2.1. Participants

In this study, a total of 10 registered male professional golfers from the Korea Professional Golfers’ Association (KPGA) participated in the putting task experiment. All participants of this research had no prior experience of olfactory treatment for their putting performance, although golfers who were interested in olfactory treatment participated in this experiment. Additionally, participants with no physical constraints and no allergic symptoms to specific scents were recruited. For the convenience of experimentation and interpretation, right-handed golfers were recruited, and the lowest duration among the participants’ careers (golf experience) was 16 years. The general characteristics of the participants in this study are shown in Table 1, below.

### 2.2. Measurements

#### 2.2.1. Inhalation

Three olfactory inhalation scents (lavender, citrus, and ammonia) were purchased from an online store as commercially available products. Ammonia inhalants were prepared in the form of sports ampoules from the Ammonia Sport company (AS Company, AMPULES, San Diego, CA, USA), and citrus and lavender inhalants were prepared with commercially available aroma oils (Edens Garden lavender and lemon essential oils, San Clemente, CA, USA). Due to COVID-19, the olfactory inhalant treatment was sprayed (in the same amounts) on the front of each participant’s mask, so that the scent could be inhaled. The mask was changed for each scent treatment, then, once the putting task was performed after each inhalant treatment, they took a break for 30 min. During the break times, all participants changed masks after washing their noses with water in the restroom. Then, the participants sat down in the waiting room next to the laboratory for relaxation purposes.

#### 2.2.2. Putting Performance

In order to perform the 3-meter putting task, a putting mat (400 cm × 50 cm) was installed in the laboratory. The participants used the same experimental putter (Tayler Made, Carlsbad, CA, USA), and the golf ball was also the same for each participant (Acushnet Company, Titlist Pro v1, Fairhaven, MA, USA). The putting task was performed eight times for each condition (control, lavender, citrus, and ammonia), and the participant’s success or failure was recorded a total of 32 times for each putting task performed by every individual. In order to ensure that the preparation time before putting commenced was different for each golfer, the participants were instructed to perform according to the signals “Ready” and “Start”. Additionally, to measure the CoP area, participants performed the putting task while barefoot on the force plate. The experimental environment for performing the putting task is shown in Figure 1 below. Although there were differences for every participant, each putting task for one inhalation treatment usually took about 6–7 min (minimum 5′ 28″, maximum 8′ 22″).

#### 2.2.3. CoP Area

A single-force platform (Kistler, Model no. 9620AA, Kistler Instrumente AG, Winterthur, Switzerland) was used to investigate the displacement of body CoP. The CoP motion in the AP and the ML directions was calculated by three force (Fx, Fy, and Fz) and moment (Mx, My, and Mz) components [32]. In order to investigate the amount of postural sway, the CoP ellipse area (cm^2^) was calculated by principal component analysis (PCA), fitting an ellipse on the area and capturing 85.35% of the CoP motion [33]. This has the advantage of estimating the amount of CoP motion, including both AP and ML directions, simultaneously [34]. The movement of the putter was recorded using a three-dimensional motion analysis system (Vicon Motion Systems, T10S, Oxford, UK) that detects the motion of a passive marker that is placed on the top of the putter head (shown in Figure 1). Eight infrared cameras were uniformly spaced and hung from the ceiling to capture the marker; based on the marker’s position, the whole task was divided into three phases, namely, the address, takeaway (from the address to the top of the backswing), and downswing (top of the backswing to the end of the swing) for further data analyses. The force platform and the motion analysis system were synchronized for data collection and were sampled at 100 Hz. The raw data sampled by both devices were filtered by a low-pass, fourth-order Butterworth filter with a cutoff frequency at 5 Hz.

#### 2.2.4. Heart Rate

The changes in HR following olfactory treatment were recorded using the HEXOSKIN biometric shirt (Hexoskin smart shirt, Hexoskin, Montreal, QC, Canada) during the putting task. The biometric shirt is Bluetooth-linked with a tablet application that calculates the average, maximum HR, and respiratory volume in real time. Hence, it is effective in observing changes in HR according to the three types of olfactory inhalation scents. During the break, the participants were wearing biometric shirts and checked whether the sensors were working. We tried to investigate the level of arousal through the HR changes collected using the biometric shirt.

### 2.3. Procedure

Before conducting the experiment, meetings with experts were conducted to review the experimental research design and verify the devices. Additionally, this research was approved by the Institutional Review Board (IRB) of Seoul National University, Korea (approval number: 2004/002-005). Next, KPGA golfers were recruited through the recruitment document, and 10 golfers participated in this research. Prior to the experiment, participants were informed about the devices, the three olfactory inhalant scents, and about measurement using the force platform.

Each participant performed the putting tasks eight times for each condition: control (no inhalation), lavender, citrus, and ammonia inhalation. According to the Latin square design [35], three groups (4 participants, 3 participants, 3 participants) were divided and treated differently according to the order of inhalant scent treatment: control–lavender–citrus–ammonia; control–citrus–ammonia–lavender; control–ammonia–lavender–citrus. 

### 2.4. Data Analysis

HR changes and putting success were summarized in an Excel file, and the CoP signals obtained during the putting task experiment were processed using a customized MATLAB (Mathworks, Natick, MA, USA) and then analyzed. A descriptive statistical analysis was performed for each inhalant scent treatment (no inhalation, lavender, citrus, and ammonia) through the organized data. Additionally, to verify the differences between groups according to treatment environment, a homogeneity of variance test, one-way analysis of variance (ANOVA), and post hoc analysis (Scheffe) were performed, using SPSS (ver. 21.0, IBM, Armonk, NY, USA). Through the eta-square values, the effect size of the olfactory treatment for each phase of putting movement was confirmed. The level of statistical significance was set at 0.05.

## 3. Results

### 3.1. Putting Performance According to Olfactory Inhalation

To verify the effect of olfactory inhalation on putting performance between the groups, a one-way ANOVA was performed. The changes in putting performance after olfactory inhalation are shown in Table 2, below. As a result, the average putting success rate according to the olfactory inhalation treatment was the highest in the control group without inhalation treatment (68.75%), and the lowest success rate was found in the citrus group (40%). However, there were no significant differences between the types of inhalation scents (*F* = 2.115; *p* = 0.115).

### 3.2. CoP Area According to Olfactory Inhalation

The changes in CoP area as a result of olfactory inhalation treatment are shown in Table 3, below. To analyze the CoP area, we filtered the data by dividing each participant’s putting motion into three phases: address, takeaway, and downswing. As shown in Table 3, the olfactory inhalation treatment was found to have statistically significant effects on CoP area changes, except for the takeaway phase during putting. In the address and takeaway phases, the CoP area values were high for the lavender scent treatment, and high in the downswing and the total for the putting phase; the CoP area value was high for the citrus scent treatment.

First, in the address phase, the mean value across the participants increased with olfactory inhalation treatment for the CoP area. There was a significant difference according to the types of inhalation scents. The results revealed that the olfactory inhalation treatment inhibited the postural stability compared to the control group in the address phase (*F* = 13.213; *p* = 0.000). Additionally, in the downswing phase, citrus inhalation was found to inhibit postural stability, compared to the other conditions (*F* = 6.423; *p* < 0.349). Similarly, in the total for the putting phase, the results revealed that citrus inhalation inhibited postural stability compared to other conditions (*F* = 5.085; *p* = 0.047).

Additionally, the effect size of the olfactory treatment on CoP for each stage of the putting task was confirmed through the eta square value, and the effect size of inhalation on the CoP area was large (eta-squared > 0.014). The results confirmed that inhalation has a greater effect on the address phase than on the downswing and total putting phase.

### 3.3. HR According to Olfactory Inhalation

The changes in HR monitored throughout the olfactory inhalation treatment are shown in Table 4, below. To analyze the HR, we filtered the data by dividing each participant’s putting motion into three phases: address, putting, and the total putting phase. In the results, the average values for the HR, sorted according to the olfactory inhalation treatment, showed a similar level regardless of the inhalation treatment in all putting task phases. As shown in Table 4, as a result, there was no statistically significant difference in HR changes for olfactory inhalation treatment according to the conditions of inhalation scents (total putting phase: *F* = 0.526, *p* = 0.666).

## 4. Discussion

This study focused on understanding the effects of diverse various olfactory inhalation treatments for trained professional golfers. Through this study, we hoped to ascertain the appropriateness of an olfactory inhalant treatment for sports that require a stable posture, rather than sports that require a high arousal level, such as weightlifting, American football, ice hockey, etc. Hence, we tried to interpret the changes in putting performance and responses (CoP area, HR) for each of the scents.

First, the results for putting performance seem to show the same results as in Pelz’s previous study [9], which had a low putting success rate according to distance. This result suggests that, despite the participants being professional golfers, the burden of having to succeed with a putting distance of 3 m and the unfamiliar experimental environment other than the green may have hindered their performance. In addition, although this study is meaningful in that it confirmed the response without adaptation to olfactory inhalation treatment, the scenario may also have acted as an obstacle to automated putting movement. To establish this, it is necessary to verify the change via adaptation to olfactory inhalation treatment and repeated measurement in follow-up studies. Additionally, it has been reported that the lower CoP area, CoP excursion, and ML direction changes are related to putting performance [3,5,36]; however, only the CoP area was confirmed in this study. CoP parameters are important factors in postural stability that affect putting performance [37]. Thus, in follow-up studies, it will be necessary to simultaneously observe several CoP variables according to olfactory inhalation treatment.

Our findings confirmed the difference in static stability as a result of olfactory inhalation treatment on performance in the putting task. A previous study on static stability through the same olfactory inhalation treatment showed that citrus and ammonia scents had a positive effect on static stability, whereas the lavender scent inhibited static stability [24]. However, in the experimental application shown in this golf putting task, the olfactory inhalant treatment was found to inhibit putting performance overall, as well as the dynamic stability of small movements, showing that olfactory inhalation treatment is not appropriate for professional golfers. Therefore, indiscriminate olfactory inhalation treatment when performing fine motor control techniques should be avoided as it inhibits dynamic stability. In addition, through the results of this study, it was verified that there is a clear difference compared to contact sports in which ammonia is inhaled with good effect during weightlifting or ice hockey, sports that require high muscle strength and a high level of arousal.

In previous studies, olfactory inhalation treatments were mainly applied to muscle strength tasks such as Sargent jump performance, etc., but in this study, CoP variables and HR changes were examined along with task performance in professional golfers. Although it was shown that the olfactory inhalation treatment inhibited the putting performance and postural stability in this study, it is judged that it will be necessary to examine motion, such as the jerk value of the putter head, during task performance in any follow-up studies. In addition, in previous studies on golf putting, for the most part, only the data collected from case studies with successful performance direction were used [3,6]; however, in this study, the olfactory inhalation treatment effects were confirmed by including the data collected from unsuccessful performance in the analysis. For follow-up studies that focus on motion analysis, it is recommended that researchers should compare both success and failure performance case studies or examine performance changes.

More recent studies have reported that olfactory inhalation treatment and aromatherapy decrease trait anxiety [38] and affect the functional condition of athletes [2]. In addition, previous studies revealed the relationship between olfactory inhalation treatment and psychological benefits, as well as arousal level [39,40,41]. In fact, in sports, olfactory inhalation treatment for arousal control is applied differently, dependent upon the sport, and at various points in time, such as during physical and psychological relaxation, at rest, just before a game, and during a game. Therefore, it is necessary to consider the effective timing and applicability of olfactory inhalation to verify the approach of olfactory treatment in more diverse sports events and time points. Additionally, in this study, there was a difference in CoP area according to the type of olfactory inhalation treatment. In particular, it was found that the citrus inhalation treatment inhibited dynamic stability most strongly compared to other olfactory inhalation treatment agents throughout the putting phase. This can be interpreted as inhibiting concentration, compared to the psychologically refreshing effect of citrus in Dunning’s (2005) study [42].

The results of this study are contrary to the results of previous studies that revealed the relationship between static stability and olfactory inhalation treatment [24]. As a result of this study, it was confirmed that the optimal level of arousal required for a quiet standing and golf putting task is different from more active sports, and it is meaningful to verify the difference between static stability and dynamic stability. In conclusion, this study shows that olfactory inhalation treatments are not effective for automatic professional golfers in terms of improving dynamic stability, suggesting that indiscriminate olfactory inhalation treatments should be avoided during and immediately before performing tasks requiring delicate control and movement. Lastly, we tried to interpret the level of arousal through HR changes, but there was no significant difference between the groups in the address and putting phases. This is a limitation of our study, and it seems that the experimental situation acted as a stress to the participants. To consider this, in follow-up studies, it is necessary to present resting HR data, such as the resting average of HR, in the analysis.

## 5. Conclusions

This study revealed the effects of olfactory inhalation treatment on professional golfers’ putting performance, CoP area (postural stability), and HR. Specifically, the olfactory inhalation treatment had a statistically significant negative effect on postural stability. The results outlined show the effectiveness of applying an olfactory inhalation treatment in tasks that require fine motor control, such as golf putting. Consequently, olfactory inhalation treatment can decrease putting performance, affecting postural stability through arousal regulation. The results also suggest that it is inappropriate to apply olfactory inhalation treatment to reach an optimal level of arousal for peak putting performance. The results of this study can be used as basic data to explore the application methods of olfactory inhalation treatment for dynamic stability in fine motor control performance. Lastly, these results are expected to provide guidelines for application in numerous sports, while considering the health and safety of athletes. In addition, through this study, it is expected that the findings will contribute to helping players to achieve peak performance more stably in competitive sports situations.

## Figures and Tables

**Figure 1 ijerph-19-12666-f001:**
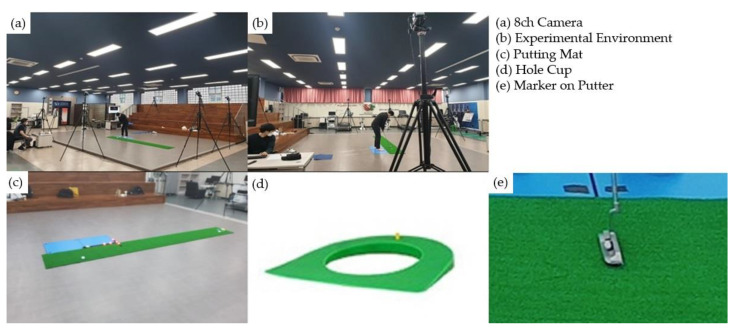
Experimental environment.

**Table 1 ijerph-19-12666-t001:** Characteristics of the research participants (*n* = 10).

Participants	Age (yrs)	Career (years)
A	33	20
B	38	23
C	34	21
D	32	20
E	28	16
F	33	23
G	30	17
H	28	17
I	31	19
J	28	17
Mean (±SD)	31.50 (±3.21)	19.30 (±2.54)

**Table 2 ijerph-19-12666-t002:** Putting performance according to olfactory inhalation (*n* = 10).

	Inhalation	Mean (%)	SD	*F*	*p*
Putting Performance (Percentage of success)	Control	68.75	20.62	2.115	0.115
	Lavender	51.25	25.98		
	Citrus	40.00	28.13		
	Ammonia	52.50	27.51		

**Table 3 ijerph-19-12666-t003:** CoP area according to olfactory inhalation (*n* = 10).

Phase	Inhalation	Mean (cm^2^)	SD	*F*	*p*	*η* ^2^	Post Hoc
Address	Control ^a^	4.69	1.80	13.213	0.000 ***	0.524	^a^ < ^b^, ^d^ *** ^a^ < ^c^ **
	Lavender ^b^	13.48	3.17				
	Citrus ^c^	11.58	4.84				
	Ammonia ^d^	13.30	3.87				
Takeaway	Control	976.16	461.89	1.088	0.367	-	-
	Lavender	1004.08	592.48				
	Citrus	726.02	279.48				
	Ammonia	769.13	307.87				
Downswing	Control ^a^	4120.61	1350.84	6.432	0.035 *	0.349	^c^ > ^a^, ^d^ * ^c^ > ^b^ **
	Lavender ^b^	3895.50	1224.84				
	Citrus ^c^	6272.46	2176.72				
	Ammonia ^d^	4006.51	1309.68				
Total	Control ^a^	1700.49	129.95	5.085	0.047 *	0.298	^c^ > ^a^, ^b^, ^c^ *
	Lavender ^b^	1637.69	89.55				
	Citrus ^c^	3003.35	413.69				
	Ammonia ^d^	1596.31	99.65				

* *p* < 0.05; ** *p* < 0.01; *** *p* < 0.001; Scheffe’s post hoc test was performed.

**Table 4 ijerph-19-12666-t004:** Heart rate according to olfactory inhalation (*n* = 10).

Phase	Inhalation	Mean	SD	*F*	*p*
Address	Control	96.22	11.92	0.282	0.838
	Lavender	95.37	12.57		
	Citrus	94.72	13.58		
	Ammonia	94.43	12.17		
Putting	Control	93.80	13.33	0.226	0.878
	Lavender	93.57	13.46		
	Citrus	92.71	14.22		
	Ammonia	92.37	12.91		
Total	Control	95.01	12.37	0.526	0.666
	Lavender	94.47	12.71		
	Citrus	93.72	13.57		
	Ammonia	90.41	12.25		

## Data Availability

The data are available upon requested from corresponding author.

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
