# Peer review of "The Effect of Olfactory Inhalation on KPGA Golfers’ Putting Performance, Postural Stability and Heart Rate"

_ijerph, 2022, doi:10.3390/ijerph191912666_

Round 1
Reviewer 1 Report
the study is of considerable interest and current. the possibility of intervention with harmotherapy in psychological and psychopathological areas is not recent but has been in use for several decades, the particularity in. this case is the application in a sport of dexterity, attention and balance such as golf. good research with careful and precise human recruitment, good application of inhalations and laboratory experimentation to standardize research as much as possible and create repeatability requirements. excellent data collection and interpretation of the same according to strict data processing canons which gives the study a good scientific character. the results are interesting both for the scientific community that will continue the study for specific sports in improving performance, balance and posture, and for athletes who will be able to grow and improve their records.
Reviewer 2 Report
Ahn & Ko present information that seemingly contribute meaningfully to the body of literature regarding olfactory stimulation-mediated athletic performance. Although I believe the methods were appropriate and largely comprehensive in answering the research question, there are several grammatical flaws and points of clarification that are required to be made before I can recommend this manuscript for publication once re-evaluated.
General Comments:
Although the flow of the paper is appropriate, there are several instances where overall grammar can be improved. I would recommend that if English is not the first language of the authors, that an external source is brought in for proofreading purposes to enhance readability of the current manuscript. If this recommendation is not taken, I have also included a few of the more abrasive examples below in the specific comments section.
Abstract:
The abstract requires both more background information preceding the purpose of the study and numerical data for readers to takeaway more than just the authors’ conclusions/interpretations.
Introduction:
The rationale to include ammonia is sound, however, there is insufficient background information with regards to lavender and essentially nothing for citrus. Additionally, the hypotheses need more clear directions (i.e. general differences are vague). Either assume the null if the question is more novel, or use background information to infer a direction.
Methods:
In the many instances where company accreditation is mentioned, please also include the state and not just generally mention the country.
I’m curious if there were any specific inclusion and exclusion criteria. I believe it is important to know whether the athletes in this study have any prior experience with olfactory inhalants for performance purposes.
In the data analysis section, please include whether normality and homogeneity of variance was tested. Furthermore, the strength of all included data would be improved if you included some form of effect size metric. For instance, your statistical model would suggest partial eta squared would be appropriate.
Results:
Perhaps I was reading the description incorrectly, but I am confused about how outcome measures varied from one another as they are described in the text versus in tables. For example, the first paragraph of the results would suggest putting performance “decreased with olfactory inhalation for control condition”, but the subsequent sentence indicates that citrus decreased by 40%. The next sentence clears things up quite a bit when discussing equivocal statistical findings, so I would recommend cutting out any statements of increase or decrease unless there were statistically significant findings. There are multiple confusing clusters of sentences in the results that could be amended this way (i.e. Page 6, Lines 216-221)
Since your statistical methods are consistent, there is no need to reiterate that the tests performed were One-way ANOVA with Sheffe post-hoc tests.
Specific Comments:
Page 1, Line 13: “…while a force plate…”
Page 1, Line 14: I don’t believe that stating the statistical model and post-hoc tests in the abstract is necessary beyond a conference abstract. Hence, it is wasted space here.
Page 1, Lines 21-23: Combine these sentences.
Page 1, Lines 35-46: This sentence is awkward and needs to be rewritten.
Page 1, Line 38: Is “Putting Perform” a proper noun or was this intended to be “Putting performance”?
Page 2, Line 48: “…to 25% at a 10 foot distance.”
Page 2, Line 64: “… that requires fine motor patterns…”
Page 2, Line 69: “…reported as a component of the nervous system that responds…”.
Page 2, Line 70: “…powerlifters”.
Page 2, Line 71: “…during the events.”
Page 2, Lines 70-74: Combine these sentences.
Page 2, Lines 74-78: “Recently, many studies have been conducted to explain the relationship between actual performance and olfactory inhalation treatment, including both general and handgrip strength improvements, Sargent jump and cardiovascular enhancements, as well as isometric mid-thigh pull velocity augmentations.”
Page 2, Lines 87-88: This sentence is awkward. Please rephrase.
Page 3, Table 1: Is the career length listed a total career of playing golf or specifically the subjects’ professional careers.
Page 3. Line 117: What is a “refreshing break” and what was the range of time? More than 30 minutes could be 31 minutes or 3 hours, etc.
Page 3, Line 124: Although it is mentioned later in the paper, listing that there are three inhalant conditions and then stating a total of 32 observations is confusing. Please describe the control condition earlier in the methods and specifically indicate what this condition entailed. I am also confused how long each subject was exposed to each condition, i.e. if they had citrus in/on their mask, were they smelling that the entire putting task? What is the general amount of time per condition?
Page 3, Line 129: Figure 1 should be listed after this line and not on the next page.
Page 4, Line 140: It would greatly benefit methodology understanding if a figure was added that demonstrates where the passive marker was placed on the putter head.
Page 4, Line 151: Is the HEXOSKIN biometric shirt a validated piece of equipment? If not, that could be a major red flag for the reliability of any heart rate data obtained via this method.
Page 4, Lines 159-162: While important, I believe that this information is a given if you have IRB approval.
Page 4, Line 167: Once again, please clarify what your control condition consisted of. I am unsure if control was simply no inhalant-mediated olfactory stimulation.
Page 5, Lines 182-183: This sentence is redundant and should thus be removed.
Page 5, Line 207: What was the specific p-value?
Page 6, Table 3: The asterisks are helpful, but it would be helpful to see the actual p-values.
Page 7, Lines 243-245: Do you mean longitudinal studies are needed? I’m slightly confused as to what your suggestion for future research is.
Page 7, Lines 247-249: These sentences on lavender are valuable background that should have been included in the introduction.
Page 7, Lines 254-256: You emphasize the difference between fine motor control (i.e. “small movements”; please change vernacular) and events that may emphasize more gross motor control and commensurate stimulation/psychological arousal. What is missing is an interpretation of why those sports are physiologically different from each other and why some might benefit from olfactory stimulation and others may not.
Page 7, Line 269: “More recent studies have reported that olfactory…”
Page 7, Line 274: What is “stability” in this context?
Page 7, Line 278: Once again, what is dynamic stability of citrus in this context?
Page 8, Line 279: What is the “refresh effect” of citrus?
Page 8, Line 286: “…treatments control dynamic stability…”.
Reviewer 3 Report
- pay attention to english grammar and syntax rules all over the manuscript
- be careful to acronyms and tables footnotes
- lines 89-94 go before the aim of the study, and not after: you could also delete it, 'cause you could talk about it in the discussion
- clearly state inclusion and exclusion criteria
- Table 1 goes in the results section
- pay attention: statistic details don't go in result section!
- be more coincise in the conclusions!
- please improve final paragraphs of your article by giving some possible next applications of your findings: i.e. how olfactory could affect other sports (gymnastics - https://link.springer.com/article/10.1007/s11332-020-00713-8) or could integrate other kinds of training (i.e. reaction time)
Round 2
Reviewer 2 Report
Ahn & Ko have improved upon the first iteration of their investigation on olfactory inhalation on varying metrics of professional golfer performance. Nevertheless, there are still minor revisions that encompass aspects that remain unclear despite the authors’ edits. Although my previous suggestion to improve upon English grammar was somewhat received, there are multiple instances throughout the most recent iteration that still warrant major proofreading. If neither of the authors are primarily English speakers or do not have a sufficient grasp on English grammar, I would suggest bringing in a third-party source to provide the aforementioned service. The current manuscript is succinct with respect towards content, but absolutely needs editing to improve its readability in a higher impact journal.
General Comments:
As mentioned above, there is much work to be done to clarify the readability of this manuscript before publication. I have provided line-by-line feedback for salient examples, but these suggestions should not be seen as exhaustive. It is beyond my scope as a reviewer to completely rework the grammar of a manuscript.
Abstract: A background prior to the purpose is still not present. The first line directly starts with a purpose. Furthermore, many of the p values included throughout the abstract do not have their appropriate directional markers (i.e. “=, <, >, etc.). Please fix these issues.
Introduction: Although the authors address that the question is novel, it would be more methodologically appropriate to assume the null rather than provide directional hypotheses.
Specific Comments:
Page 1, Lines 33-34: This sentence is awkward and needs to be rephrased.
Page 2, Lines 55 & 70: It is unclear whether “this” refers to the studies being discussed or the present investigation. It would be inappropriate to perform the latter since none of the methods have yet been discussed, but the wording needs to be clarified regardless.
Page 2, Line 57: “…was set to 3-meters.”
Page 2, Lines 89-91: These sentences are redundant to lines 85-88.
Page 3, Line 132: The authors contend that a more sufficient explanation of the break between conditions was supplied, but I fail to notice any changes beyond a simple rewording. The “refresh” time is still entirely vague to me and needs to be explained. Did the participants simply do whatever they wanted in the interim?
Page 5, Line 197: “…was set at p<.05”?
Page 7, Line 244, “This study focused on understanding…”.
Page 7, Line 253: “…distance of 3-meters and…”
Page 7, Line 267: “…in static stability, whereby lavender scent…”.
Page 8, Line 272: “…as it inhibits dynamic stability.”.
Page 8, Line 277: So were the previous studies not in professional golfers? Were they recreational? This is unclear.
Page 8, Line 291: …”and psychological benefits, as well as arousal level.”.
Page 8, Line 292: What does “character” mean in this instance?
Page 8, Line 297: “…CoP…”.
Page 8, Line 298: “…treatment inhibited dynamic stability…”.
Page 8, Line 318: “… inhalation treatment had a statically significant negative effect on postural stability.”
Author Response
We grateful for your time and comments on our submission [ijerph-1877526].
Our manuscript used the MDPI English editing service according to the reviewer’s comment. The appropriate changes made in the revised manuscript are yellow highlighted.
We believe that these modifications have strengthened the manuscript and hope that the revised manuscript is suitable for publication in IJERPH.
Abstract
A background prior to the purpose is still not present. The first line directly starts with a purpose. Furthermore, many of the p values included throughout the abstract do not have their appropriate directional markers (i.e. “=, <, >, etc.). Please fix these issues.
>Thank you. We have added the background paragraph in abstract section according to your comment. Also, we have supplemented the numerical notation. The modifications are all yellow highlighted.
Introduction
Although the authors address that the question is novel, it would be more methodologically appropriate to assume the null rather than provide directional hypotheses.
> Thank you for the valuable comments. We have changed to null hypotheses. It is yellow highlighted.
Specific Comments
- Page 1, Lines 33-34: This sentence is awkward and needs to be rephrased.
> We have revised that sentence. Thank you. It is yellow highlighted.
- Lines 55 & 70: It is unclear whether “this” refers to the studies being discussed or the present investigation. It would be inappropriate to perform the latter since none of the methods have yet been discussed, but the wording needs to be clarified regardless.
> Thank you for your opinion. We have omitted the unnecessary sentence of Line 55. And we have modified for clarity. The modifications are yellow highlighted.
- Page 2, Line 57: “…was set to 3-meters.”
> We have modified it. It is yellow highlighted.
- Page 2, Lines 89-91: These sentences are redundant to lines 85-88.
> We have modified it.
- Line 132: The authors contend that a more sufficient explanation of the break between conditions was supplied, but I fail to notice any changes beyond a simple rewording. The “refresh” time is still entirely vague to me and needs to be explained. Did the participants simply do whatever they wanted in the interim?
> We have supplemented the explanation of break time. It is yellow highlighted.
- Page 5, Line 197: “…was set at p<.05”?
> We have revised to ‘the level of statistical significance’. It is yellow highlighted.
- Page 7, Line 253: “…distance of 3-meters and…”
> We have modified it. It is yellow highlighted.
- Page 7, Line 267: “…in static stability, whereby lavender scent…”.
> We have modified it. Thank you.
- Page 8, Line 272: “…as it inhibits dynamic stability.”.
> We have checked it again.
- Page 8, Line 277: So were the previous studies not in professional golfers? Were they recreational? This is unclear.
> We have modified that sentence. It is yellow highlighted.
- Page 8, Line 291: …”and psychological benefits, as well as arousal level.”.
> We have modified it. Thank you.
- Page 8, Line 292: What does “character” mean in this instance?
> We have rewritten the sentence. Thank you for the comments. It is yellow highlighted.
- Page 8, Line 297: “…CoP…”.
> We have modified it.
- Page 8, Line 298: “…treatment inhibited dynamic stability…”.
> We have modified it.
- Page 8, Line 318: “… inhalation treatment had a statically significant negative effect on postural stability.”
> We have modified it. Thank you.
